# Comprehensive nutrient analysis in agricultural organic amendments through non-destructive assays using machine learning

Erick K. Towett[1]*, Lee B. Drake[2]*, Gifty E. Acquah[3], Stephan M. Haefele[3], Steve P. McGrath[3], Keith D. Shepherd[1]

1 World Agroforestry (ICRAF), Nairobi, Kenya, 2 Department of Anthropology, University of New Mexico, Albuquerque, NM, United States of America, 3 Department of Sustainable Agricultural Sciences, Rothamsted Research, Harpenden, United Kingdom

* e.towett@cgiar.org (EKT); b.lee.drake@gmail.com (LBD)

**Data Availability Statement:** The replication data that support the findings of this study and the XRF and MIR .quants are available from the World

## Abstract

Portable X-ray fluorescence (pXRF) and Diffuse Reflectance Fourier Transformed Mid-Infrared (DRIFT-MIR) spectroscopy are rapid and cost-effective analytical tools for material characterization. Here, we provide an assessment of these methods for the analysis of total Carbon, Nitrogen and total elemental composition of multiple elements in organic amendments. We developed machine learning methods to rapidly quantify the concentrations of macro- and micronutrient elements present in the samples and propose a novel system for the quality assessment of organic amendments. Two types of machine learning methods, forest regression and extreme gradient boosting, were used with data from both pXRF and DRIFT-MIR spectroscopy. Cross-validation trials were run to evaluate generalizability of models produced on each instrument. Both methods demonstrated similar broad capabilities in estimating nutrients using machine learning, with pXRF being suitable for nutrients and contaminants. The results make portable spectrometry in combination with machine learning a scalable solution to provide comprehensive nutrient analysis for organic amendments.

## Introduction

Small-scale farmers produce 80% of the food supply in developing countries, and investments to improve their productivity are urgently needed [1]. Achieving higher rates of productivity will need to rely on improved technologies such as high-yielding crop varieties, better and more inorganic fertilizer, and more efficient use of available resources, for example manures and other organic fertilizers. Effective quality assurance mechanisms can help address three challenges facing scaling-up efforts in supply chains for foods in developing countries of Sub-Saharan Africa (SSA): sourcing, market size and consumer trust [2]. Clark and Hobbs [2] performed supply chain analysis to evaluate how stakeholder actions and relationships influence the dynamics of complementary food markets in SSA and argued that effective signalling of

Agroforestry - Research Data Repository https://doi.org/10.34725/DVN/YTJTZQ. Spectra are saved in these files in RDS format and they work with open-source code repositories (CloudCal and cloudFTIR, both open github projects). The CloudFTIR can be found at: https://github.com/leedrake5/cloudFTIR.

**Funding:** This work received support from the Bill & Melinda Gates Foundation (BMGF), through the Africa Soil Information Service (AfSIS), and the CGIAR Research Program on Water, Land and Ecosystems (WLE), supported by the CGIAR Trust Fund: wle.cgiar.org/donors. SPM and SH were funded by the Institute Strategic Programme grant, "Soils to Nutrition" UK-BBSRC grant number BBS/E/C/000I0310. The funders were not involved in the study design, the collection, management, analysis, and interpretation of data, the writing of the report or the decision to submit the report for publication.

**Competing interests:** The authors have declared that no competing interests exist.

credence attributes via credible quality assurance can contribute to the sustainability of local complementary food supply chains and once established, may contribute to the long-term affordability, accessibility and availability of these foods in SSA. Establishing regional and/or country-level quality assurance mechanisms for agro-inputs (e.g. organic fertilizers) quality requires the coordination of stakeholder actions to address food insecurity. This is particularly important for subsistence farmers, for whom organic farming is often the only available option for at least part of their farm. Inorganic fertilizers are expensive for poor farmers [3], and their use is often restricted by cash flow problems. When small farmers do use inorganic fertilizer, it often is applied in pockets or provided to specific plants [4], or to cash crops such as cotton [5]. Hence, organic amendments (OA) such as cattle manure and mixed farmyard manure are necessary to replenish nutrients in the majority of smallholder plots. Consequently, organic amendments have been found to be essential component of strategies for integrated soil fertility management to maintain soil nutrients in both mixed crop and livestock agriculture, which are common across smallholder farmers [6–9].

Organic amendments comprise a variety of plant-derived materials that range from dried plant materials, to animal manures and litters, and agricultural by-products and sewage sludges. The nutrient content of such organic amendments varies greatly among source materials but is usually substantially lower in organic fertilizers as compared with chemical fertilizers. However, organic amendments contain macro- and micro-nutrients, and may provide added value compared with standard mineral fertilisers [10]. Organic amendments also improve soil structure, increase water holding capacity and promote biological activity [11], but what is less clear from published evidence is the relationship between an action to improve soil structure (for example addition of OA to soil) and the magnitude of change in the associated benefit (for example increase in soil carbon). But despite these potential benefits, OA may be unbalanced in terms of relative availability of nutrients [10].

To achieve an efficient combination of organic and inorganic fertilizers as well as the adjustment to crops and soils, knowledge of the composition of the OA is essential. Traditional laboratory methods for analyzing major and trace elements in OA use a combination of homogenization, drying and preparation of individual samples followed by acid digestion and determination of elements by atomic absorption spectrophotometry (AAS), inductively coupled plasma optical emission spectrometry or mass spectroscopy (ICP-OES or ICP-MS). Total carbon and nitrogen need a separate analysis, today usually done by routine dynamic combustion methods on an elemental analyzer. While these methods are well established, the associated sample preparation procedures are time-consuming and expensive. As such, they are not scalable solutions to assist subsistence farmers in developing countries. New cheap and fast analytical methods are therefore needed to enable smart nutrient resource management.

The commercial availability of portable Energy Dispersive X-ray Fluorescence (EDXRF) has enabled wider use of non-destructive total element analysis in multiple material types [12]. The range of elements which can be determined is limited by the sensitivity of the detector and its energy range; for portable EDXRF the range is from sodium (atomic number 11) to uranium (atomic number 92). The data obtained are counts of photons emitted at different energy levels and need to be calibrated to provide quantitative values [13]. This is often done using fundamental physical parameters, which employ any number of equations to estimate elemental concentrations in analytes [14]. This approach functions well for metals [15] but runs into difficulty when applied to oxides, where different oxides and carbonates can frustrate quantification [16]. However, when appropriately calibrated, portable XRF (pXRF) systems give results at the μg/g level [17]. In some cases, portable equipment is comparable to laboratory systems [18] when calibrated empirically. The quality standard for laboratory and portable equipment is based on both validity and reliability (Fig 1) [19].

Testing XRF to quantify a wide range of elemental contents in a range of organic fertilizers and their effect on crop produce could accelerate the assessment of the suitability of new fertilizer products and a variety of OA. Recently, Sapkota et al. [20] indicated that elemental concentration can accurately be measured in dried and moist manure samples using pXRF. Likewise, Roa-Espinosa et al. [21] found that XRF spectrometry can be used as a rapid and precise method for quantitative elemental analysis of macro- and micronutrients in dairy manure. Thomas et al. [22] also demonstrated that pXRF is a reliable, cost-effective tool for screening potential organic fertilizers and their effect on grains and crop residues.

Diffuse Reflectance Spectroscopy (DRS) is another method emerging as a rapid and cost-effective alternative to routine laboratory analysis for multiple sample matrices such as soils, plants and manures based on the interaction of electromagnetic energy with matter. It is already well established that mid-infrared spectroscopy (range: 4000–400 cm$^{-1}$) (MIR) is useful for estimating a number of chemical and physical soil properties (such as pH, cation exchange capacity (CEC), carbonate content, organic carbon content, soil texture, mineral composition, organic matter and water content (hydration, hygroscopic and free pore water, etc.) from a single soil spectrum with minimal sample preparation [23]. Examples of applications of infrared spectroscopy in agricultural inputs (manures/compost/bio-wastes/litter/organic resource quality) include the analyses of the following properties: Moisture, pH, total N, $NH_4$-N, total dissolved N, suspended N, soluble reactive, K, Ca, Mg, total dissolved P, suspended C, P, Ca, Na, and Mg; various salts and metals; lignin, total soluble polyphenols, decomposition rate, in vitro dry matter digestibility, C and N mineralisation potential, compost maturity [23]. Spectral absorbances from the MIR range can be calibrated using regression models to predict multiple constituents in seconds in a wide range of materials and the accuracy of these regression models relies heavily on the calibration dataset used. Although MIR spectroscopy has so far had limited use in developing countries, it has potential to make a huge contribution in helping these countries accelerate agricultural development while safeguarding their environment in the drive towards achieving the sustainable development goals [23].

A major challenge for operationalising MIR and pXRF is how to provide robust calibrations that hold up over a wide range of materials and across instruments. Solutions are needed to lay the foundations for a comprehensive nutrient measurement system that would require minimum sample preparation and meets the needs of validity and replicability [19]. This paper aimed to test machine learning methods for calibrating pXRF and MIR, both independently and combined, for analysis of macronutrient content and potential contaminants of a wide range of organic amendments. Specific sub-objective here were to examine the inter-instrument variability of six pXRF instruments and conduct an assessment of Diffuse Reflectance Fourier Transformed Mid-Infrared (DRIFT-MIR) spectroscopy for OA analysis by utilizing the MIR spectra of OA samples. Targeted characteristics were total C and N, as well as ash and other macro and micronutrients, in parallel with the pXRF analysis of the same parameters.

## Materials and methods

Ninety-eight organic amendment samples were obtained from a wide range of sources in Western Kenya (64) and the United Kingdom (34). Samples from Kenya included manure from cattle, goats, poultry and pigs whereas samples from the UK included cattle manure, mixed farmyard manure, sewage sludge and sewage sludge compost. Conventional analysis for major- and micronutrients, as well as total carbon and ash content were conducted at Rothamsted Research (UK). Total C and total N were determined using a modified Dumas method on a Leco TruMac combustion analyser. Major and trace elements were determined using ICP-OES or ICP-MS (inductively coupled plasma optical emission spectrometry/mass

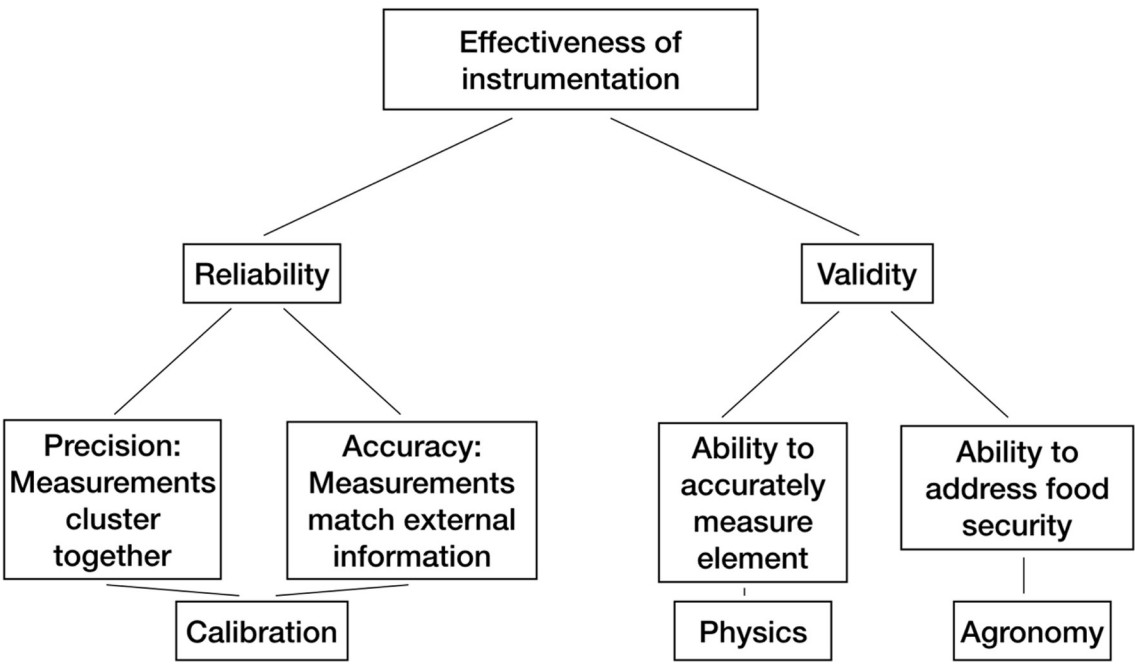

Fig 1. Core concepts in portable instrumentations, adopting the framework by Hughes (1998) [18].

spectrometry) analysis after nitric/perchloric acid (85/15 v/v) or Aqua Regia (HCl:HNO$_3$ 4:1) digestion of test samples in open tubes at up to 175 $^o$C [24, 25]. The ash content was determined using the loss-on-ignition method. Test samples that had been oven dried (105 $^o$C for at least 5 hours) were ashed in a furnace at 550 $^o$C for two hours and weighed to calculate the relative ash content (%).

Six Bruker Tracer 5i XRF instruments (900F4352, 900F4473, 900F4504, 900F4166, 900F5118, 900F5163) with Rhodium tubes were used to collect data. The units had resolutions (full width height maximum, or FWHM) of 135 eV at the Manganese K-alpha line. Samples were analyzed with a voltage of 10 kV and a current of 70 μA for 90 seconds with no filter for the elemental range of Na, Mg, P, S, K, and Ca. In addition, the samples were analyzed with 35 kV and a current of 35 μA for 90 seconds with a filter (Cu 75 μm: Ti 25 μm: Al 200 μm) for the elements Mn, and Fe. Scans were collected from air-dried and milled (to pass a 75- microns mesh sieve) samples presented as loose powder in XRF cups lined with Prolene film.

The MIR spectra of the OA were also acquired on air-dried and ground (to pass a 75- microns mesh sieve) samples using a Bruker Alpha KBr DRIFT-MIR spectrometer. On each sample holder, samples were loaded in a single replication. During scanning at each spot/sample, 32 co-added scans were collected at a resolution of 4 cm$^{-1}$. Spectra were truncated to 4000–600 cm$^1$ and regions showing atmospheric $CO_2$ features (2379.8–2350.8 cm$^{-1}$) were removed. Raw MIR spectra (Fig 2A) were transformed using the Savitzky-Golay (SG) transformation [26] with a window size of 21 data points and a polynomial order 3 [27] and followed by the first derivative transformation (Fig 2B) before developing calibration models. All MIR spectra derivations and calculations were performed with R statistical language and open source software [28]. Principal Component Analysis (PCA) was performed on the pre-processed MIR spectra of all the OA samples.

Machine learning has seen rapid advancement in the past decade owing to their implementation in open source languages such as R and python, as well as increasingly affordable high-

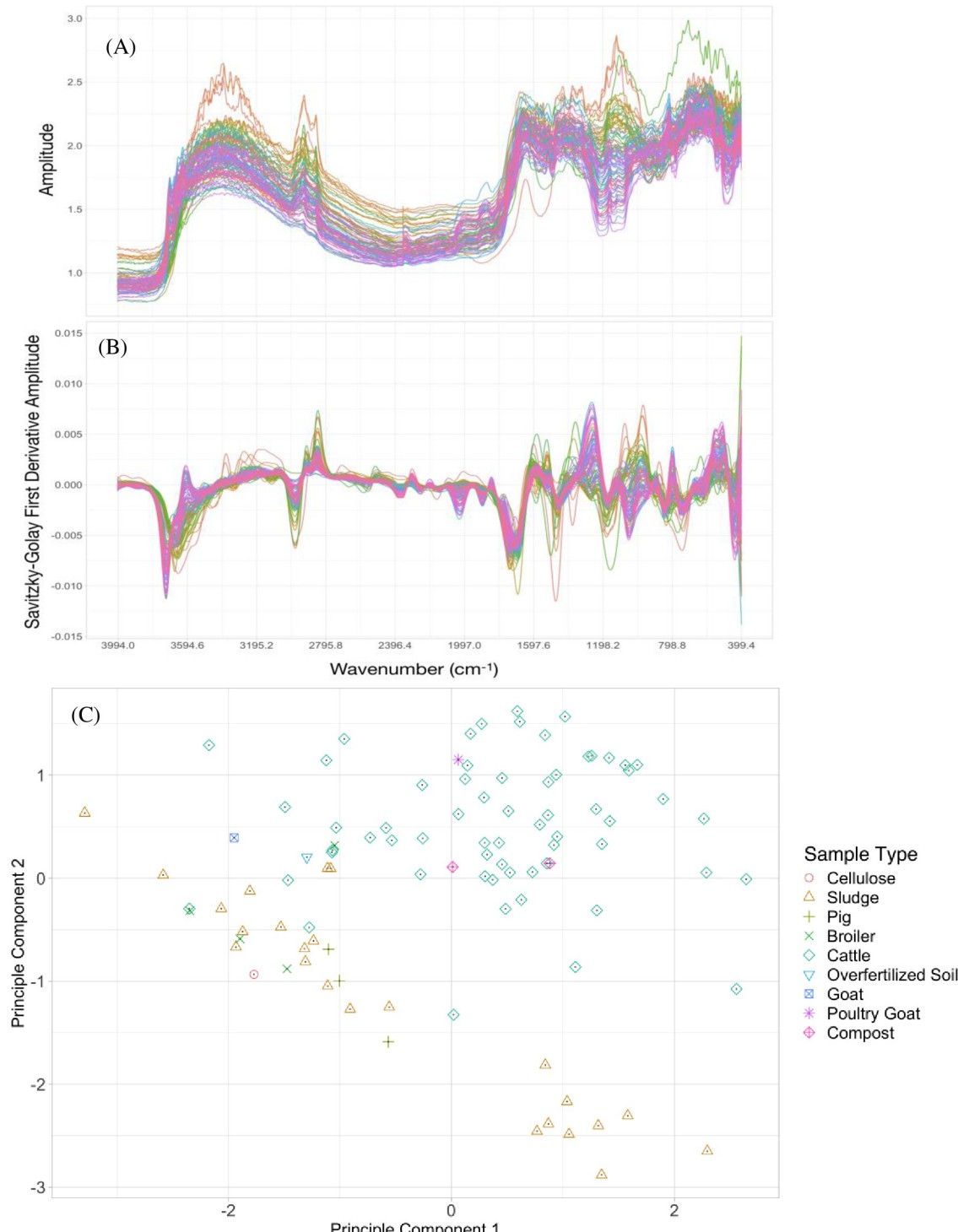

**Fig 2. Illustrations of the diffuse reflectance mid infrared (DRIFT-MIR) spectra from the organic amendment (OA) samples.**
(A) raw spectra, (B) pre-processed spectra and (C) Principal Component Analysis (PCA) scores plot (PC1 vs. PC2) for mid infrared (MIR) spectra from the OA samples (n = 98) based on sample types. The PCA was performed on the pre-processed MIR spectra.

powered processors on computers. Forest regressions are one of the simplest implementations of these techniques in which variables are randomly selected and iterated over a set number of sampling events and a set number of trees with a set number of iterations. From these, a weighted final model is produced which accounts for variable importance in a way which is generalizable (e.g. low risk of overfitting). A more advanced technique, extreme gradient boosting (XGBoost), was also used [29–31].

Typically, variables are defined via general parameters (e.g. a range of energies in a spectrum) which correspond to known responses, such as the Kα1 or Lα1 emission line for an element in pXRF. In lieu of pre-defining pXRF lines for each element in a traditional manner [32], the entire spectrum was used for machine-learning models. While it is standard practice to use the full DRIFT-MIR spectra in calibrations and variables are also defined via general parameters (e.g. a range of wavenumbers in a spectrum) which correspond to known responses, such as a functional group, it is not common for pXRF spectra. This removes the need to pre-define variables in pXRF analysis (e.g. Ca Kα1) and instead uses the whole spectrum without human input to evaluate models. This approach also allows an independent variable, such as ash content, to be quantified if suitable data is provided for training. Further, it allows a visualization of the spectrum in terms of its predictive properties for a given independent variable. Using the whole spectrum enables full automation of the calibration process for pXRF.

Forest models for pXRF and DRIFT-MIR spectra were run using 1500 trees with 200 resampling events using k-fold cross validation over 25 iterations based on the R package randomForest (4.6–14). The best forest models were selected using root mean square error (RMSE) in the caret package (6.0–84) using the R language (3.6.0). XGBoost models differ from forest in that trees can be weighted differently, have fixed depths, and different resembling. For example, trees can be built from randomly selected columns (Energies for pXRF spectra) and rows (standards). XGBoost (0.82.1) models were run using 400 rounds with a variable tree depth ranging from 5 to 25. pXRF energy channels (columns) were randomly selected with a range of 40–60%, with samples (rows) randomly selected with the same frequency. Learning rates (eta) were constrained to values between 0.1 and 0.3, with gamma regularization ranging from 0 to 0.1. The minimal child weight (controls the model complexity) was limited to 1. Unique combinations of these variables were run over 32 iterations with k-fold using caret (6.0–82) and the best model was selected using root mean square error (RMSE). Calibrations were created using CloudCal (v3) [33].

The models were evaluated using randomized cross-validation and a hold-out validation consisting of 67/33% split between calibration and validation data. This high number of standards randomly withheld from training (33%) was used to test the generalizability of models and ensure that machine learning wasn't simply memorizing data sets. Both the $R^2$ value and validation slope of the regression line between observed and predicted values for all cross-validation trials were used to evaluate the best model because (a) both metrics should approach 1 as models increase in accuracy and (b) while $R^2$ provides information regarding model precision, the validation slope gives the clearest assessment of model accuracy; a validation slope of 1 would indicate a 1:1 ratio between predicted and known values.

## Results

The wet chemistry analysis data of the nutrient values varied greatly among OA and showed the considerable diversity in the samples (Table 1). This was also confirmed by the PCA scores plot (PC1 vs. PC2) for DRIFT-MIR spectra from the OA samples (Fig 2C). The difference in the OA types is clearly shown; cattle manure seemed to group with goat and poultry manure as

**Table 1. Descriptive statistics for total carbon (C), total N, ash content, major nutrients and potential contaminants of all organic amendment (OA) samples used for the calibration and validation of pXRF and DRIFT-MIR methods.**

| Property | Units | Mean | Std dev. | Min | Max | Percentile | | | | |
| --- | --- | --- | --- | --- | --- | --- | --- | --- | --- | --- |
| | | | | | | 2.5th | 25th | 50th | 75th | 97.5th |
| Ash | % | 63.8 | 21.5 | 11.6 | 94.9 | 16.0 | 52.8 | 70.1 | 81.5 | 88.8 |
| Total C | % | 19.4 | 11.8 | 1.23 | 44.7 | 5.30 | 9.95 | 15.8 | 27.5 | 43.5 |
| Total N | % | 1.65 | 1.32 | 0.02 | 5.41 | 0.40 | 0.75 | 1.11 | 2.29 | 5.27 |
| P | % | 0.64 | 0.85 | 0.06 | 3.40 | 0.08 | 0.13 | 0.19 | 0.99 | 3.00 |
| K | % | 0.82 | 0.77 | 0.07 | 4.06 | 0.10 | 0.35 | 0.60 | 1.01 | 2.87 |
| Ca | % | 1.99 | 2.73 | 0.25 | 21.14 | 0.37 | 0.64 | 0.86 | 2.87 | 8.50 |
| Al | % | 1.95 | 1.36 | 0.01 | 8.97 | 0.02 | 0.87 | 1.84 | 2.71 | 4.28 |
| Mg | % | 0.39 | 0.34 | 0.13 | 2.79 | 0.14 | 0.24 | 0.31 | 0.43 | 1.29 |
| Na | % | 0.14 | 0.73 | 0.00 | 7.21 | 0.01 | 0.01 | 0.02 | 0.06 | 0.44 |
| S | % | 0.33 | 0.38 | 0.03 | 1.74 | 0.04 | 0.08 | 0.14 | 0.57 | 1.31 |
| Mn | % | 0.05 | 0.05 | 0.01 | 0.45 | 0.02 | 0.04 | 0.05 | 0.06 | 0.09 |
| Fe | % | 2.38 | 2.78 | 0.05 | 24.96 | 0.09 | 1.18 | 1.79 | 2.83 | 6.64 |
| Zn | mg kg$^{-1}$ | 571 | 1151 | 41 | 6553 | 43 | 63 | 92 | 432 | 3828 |
| Cu | mg kg$^{-1}$ | 278 | 762 | 10 | 5080 | 11 | 17 | 23 | 143 | 1413 |
| Ni | mg kg$^{-1}$ | 70 | 142 | 3.0 | 1015 | 3.4 | 21 | 29 | 46 | 474 |
| Cd | mg kg$^{-1}$ | 8.2 | 25 | 0.1 | 137 | 0.2 | 0.9 | 1.5 | 2.2 | 108 |
| Pb | mg kg$^{-1}$ | 129 | 310 | 0.2 | 1398 | 0.5 | 5.3 | 8.3 | 35 | 1263 |

well as compost whereas pig manure grouped together with sewage sludge. Also, there was more variance in the cattle manure group than in the sewage sludge group. The means and ranges to in elemental contents and major nutrients varied within the selected OA samples (Table 1). Nutrient concentrations ranged widely, for example 0.02–5.41% for N and 0.06–3.40% for P. Ash content as an indicator of the inorganic component ranged from 11.6–94.9%. The wide range of characteristics in the samples was also confirmed by the PCA scores plot (PC1 vs. PC2) for MIR spectra (Fig 2C).

To investigate the effect of the inter-instrumental variability on the pXRF regression models and on the prediction accuracy, calibration curves were generated for each instrument and different chemical properties. Photon data from each instrument varied, with instrument 900F4166 producing lower counts and instrument 900F4473 producing the highest (Fig 3A). For example, instrument 900F4473 had >2,500 counts per second for phosphorus K-alpha and a concentration of about 3.2% P, while instruments 900F4166 and 900F5188 had the same 2,500 counts for a much higher P concentration of >6.5%. But these differences between instruments can be corrected with instrument specific calibrations as shown in Fig 3B, where the predictions for phosphorus concentration from the different instruments overlay each other and are all close to the values measured by ICP-OES.

Next, spectroscopy (PXRF and DRIFT-MIR) in conjunction with both chemometric techniques was tested to predict various chemical properties of OA. The predictions of the calibration models compared to the measured values are shown in the Figs 4 and 5 (examples are ash and N content, respectively). For both examples, the figures show generally good predictions with little error ($R^2$ values above 0.98) and a close fit of actual and predicted concentrations (the regression line for all data points is close to the 1:1 line). For the calibrations, 67% of all OA samples were used (randomly selected). A summary of the regression coefficients for all calibrations conducted for both methods (pXRF and DRIFT-MIR), both models (Forest or XGBoost) and all OA characteristics determined is provided in Table 2. It shows that very

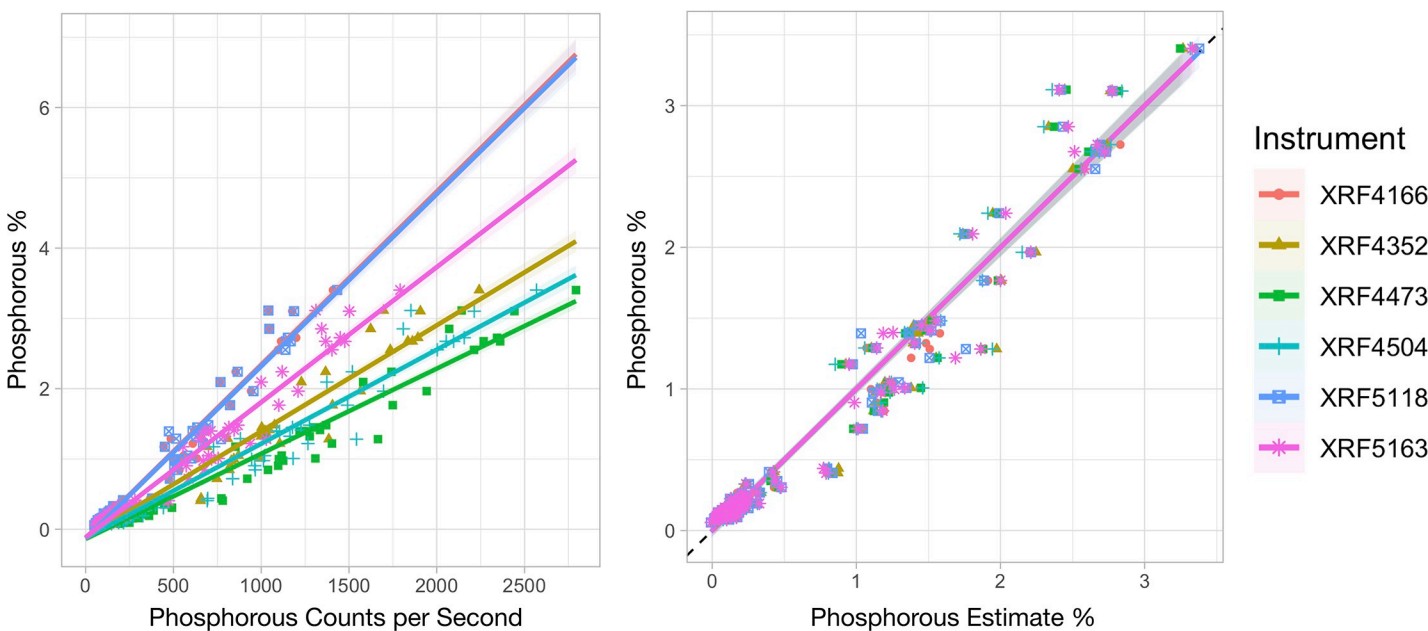

**Fig 3. Inter-instrumental variability of pXRF instruments.** Fig 3A shows the relation between phosphorus-specific counts per second for six different instruments. XRF4166 line is hidden behind the blue XRF 5118 line in the left figure. Fig 3B shows the estimated phosphorus content of the samples after instrument specific calibration. The dotted line on the right plot indicates the expected 1:1 ratio for XRF estimates and known values and the shaded areas around all lines are 95% confidence predictions based on the calibration.

good regressions were achieved for most characteristics, but they also indicate differences in the predictive value between methods and models used. Across both methods and models, good to very good regressions were achieved for ash, total C, total N, P, K, Ca, S and Fe ($R^2 >$ 0.9). Less good, but still acceptable regressions were found for the elements Mg, Na and Mn ($0.7 > R^2 > 0.9$). Against conventional knowledge, XRF performed well to predict total C, total N and ash in OA, which cannot be predicted with XRF based on known element specific $K\alpha 1$ or $L\alpha 1$ emission lines. Differences in predictive power of the calibration functions between both methods (pXRF and DRIFT-MIR) and for the characteristics tested were small in most cases. Only in the case of Na at higher concentrations did both methods underestimate the actual concentration (S2 Fig in S1 File).

Next, we used the calibration functions established with two thirds of the total sample number and tested their predictive power with the remaining one third of the samples (validation). Validation results are shown in Fig 6 for the six different pXRF instruments and four selected characteristics (ash, total C, total N, P). The regression lines indicate relatively good predictions for these "unknown samples" with a decreasing precision in the order of P > C > N > ash. In addition, the regression line for the pXRF predictions were in all four cases close to the expected 1:1 ratio between observed and predicted values. The same validation was conducted for DRIFT-MIR (Fig 7) and precision of predictions decreased in the order of N < ash < C <P. Again, the regression line for the predicted versus observed values was close to the expected 1:1 ratio.

A summary of all regression coefficients and slopes of the regression line between observed and predicted values for all cross-validation trials with pXRF and MIR and all sample characteristics is shown in Table 3. The results indicate, that the pXRF method allows generally better predictions than the DRIFT-MIR method for most OA characteristics, the exceptions are ash and carbon. Which model (Forest or XGBoost) gives the best result, based on a mixed

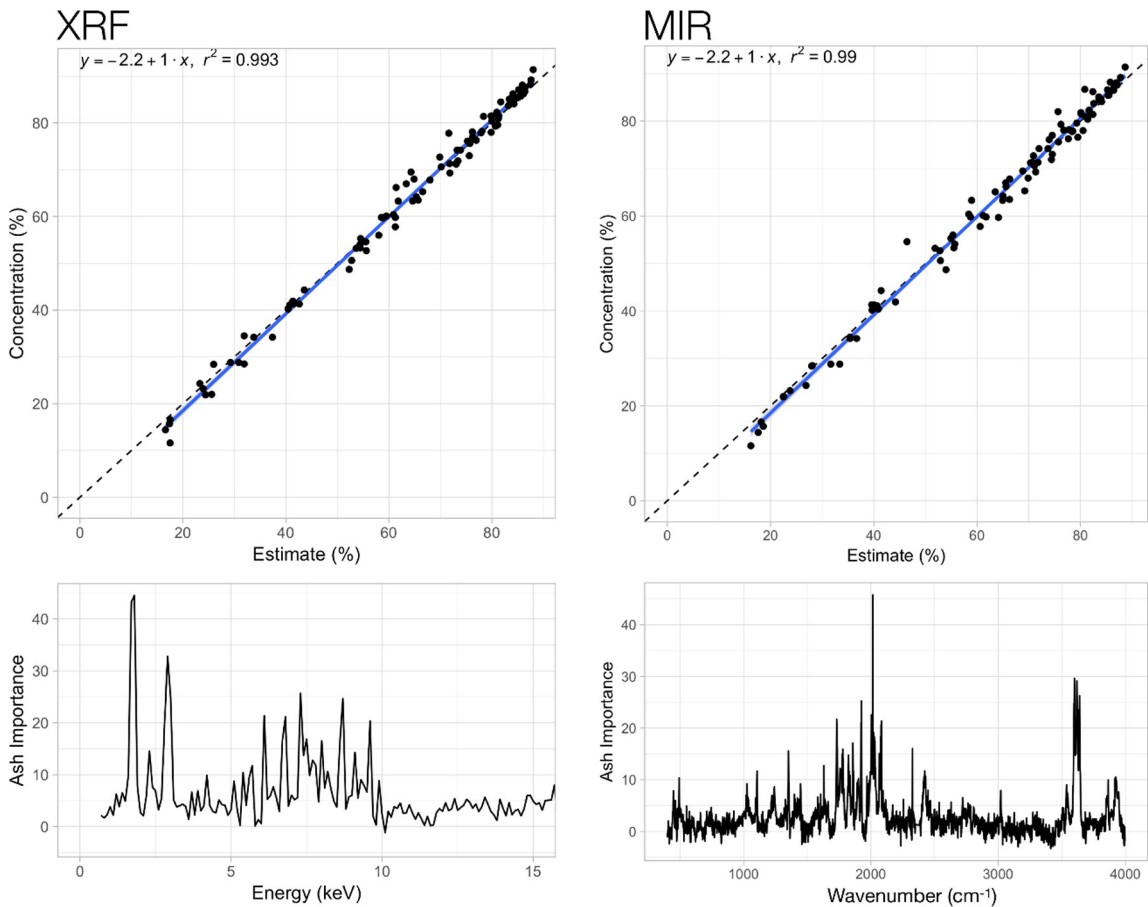

**Fig 4. Comparison of actual and predicted ash content of organic amendment (OA) samples based on whole-spectrum forest regressions for pXRF (instrument 900F4473) and DRIFT-MIR.** The dotted line indicates the expected 1:1 ratio for estimates and known values. The bottom graphs show the respective response variables in the spectra.

indicator of regression coefficient and slope of the regression line between observed and predicted values, varies between the OA properties.

## Discussion

The wide range of materials and element values for calibration and validation was helpful to i) test the potential of the methods and ii) provide robust calibrations. The principal component analysis (Fig 2C) indicated a structural difference between solid OA (manure from cattle, goat, poultry but also compost) and more liquid OA like pig manure and sewage sludge. A wider variety in components of solid OA might be responsible for a larger variance in the cattle manure group than in the sewage sludge group. We also found that spectra of OA (S1 Fig in S1 File) has some resemblance to soil spectra because there is often some soil mixed in with the manure and some soil features were therefore evident (e.g., O-H stretching in clays at 3694 cm$^1$). This similarity occurs even though most soil organic matter derives from the decomposition of plant material added to the soil whereas most organic matter in manure comes from partially digested plant material eaten by the animals. This finding was also in agreement with that of a previous study [34] that found that the quality of most of the manure resources derived from cattle, sheep, goat, chicken in selected household across four district in a semi-

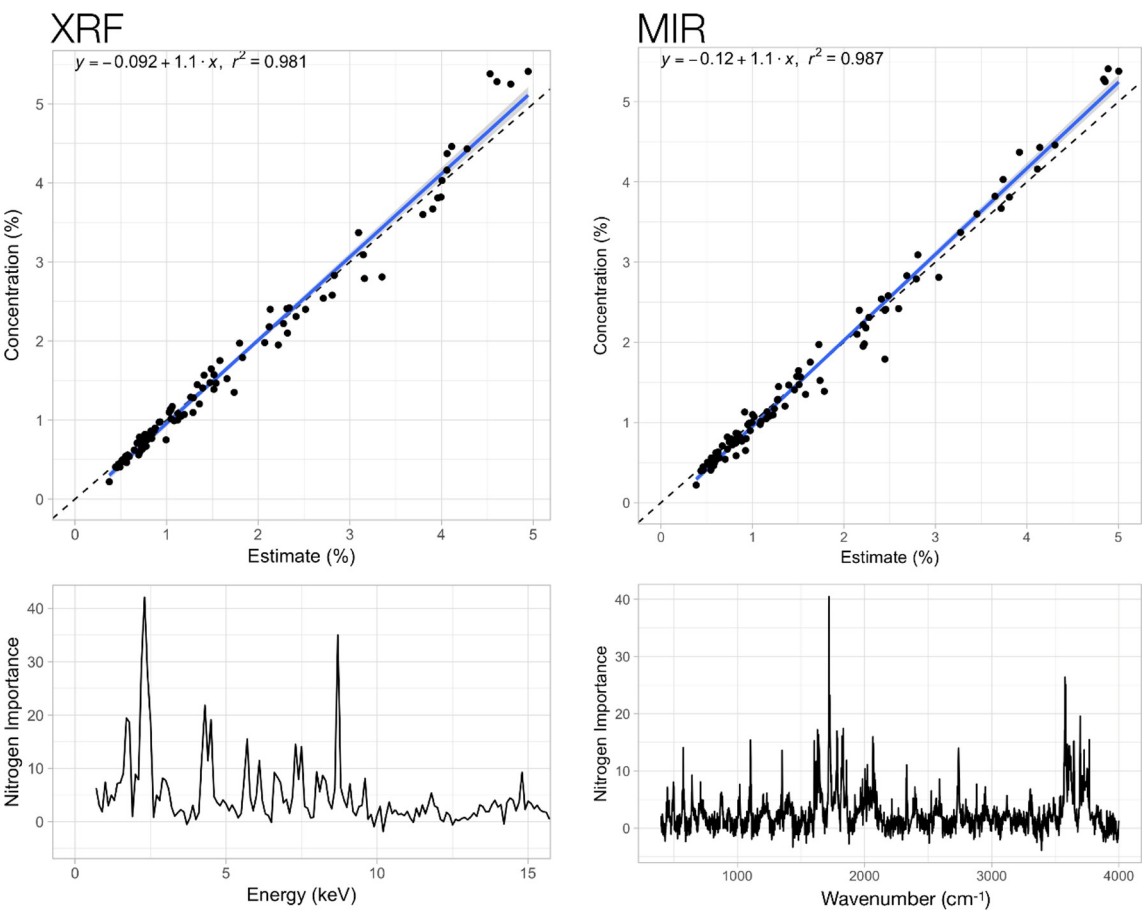

**Fig 5. Comparison of actual and predicted total nitrogen (N) content of organic amendment (OA) samples based on whole-spectrum forest regressions for pXRF (instrument 900F4437) and DRIFT-MIR.** The dotted line indicates the expected 1:1 ratio for estimates and known values. The bottom graphs show the respective response variables in the spectra.

arid environment of the North West Province in South Africa had relatively high soil content (mean 22.7%).

Using this diverse set of samples, we explored the possibility of a comprehensive analysis of OA, employing machine learning methods to evaluate the MIR and pXRF spectra, namely random forest regressions and extreme gradient boosting. The emergence of these MIR and pXRF systems presents new opportunities for rapid, low-cost analysis of OA samples, both as lab system and portable systems. We hypothesized that pXRF instruments could provide OA data of sufficient accuracy and would reduce the overall time and budget compared with the use of conventional techniques. However, their sensitivity and accuracy are dependent on the instruments' settings, make and model [35]. We found that the photon data from the various Tracer 5i pXRF instruments varied considerably, with 900F4166 producing much lower counts than the 900F4473 instrument (Fig 3A). This variation results from small changes in the anode thickness of the X-ray tube and imperfections in tube-sample-detector geometry. Therefore, each individual pXRF instrument needs to be calibrated separately.

Our assessment of pXRF and DRIFT-MIR spectroscopy for the analysis of total C and total N as well as total elemental composition of multiple elements in OA samples confirmed the potential of these tools. Using forest regression and extreme gradient boosting machine learning models, excellent calibration functions could be established (Table 2; Figs 4 and 5) to

**Table 2. Correlation ($R^2$) values for pXRF and MIR calibrations for total C, total N ash content, and major nutrients.**

|  | pXRF Forest $R^2$ | pXRF XGBoost $R^2$ | MIR Forest $R^2$ | MIR XGBoost $R^2$ |
|---|---|---|---|---|
| Ash (%) | 0.94 | 0.94 | 0.93 | 0.94 |
| Total C (%) | 0.97 | 0.92 | 0.95 | 0.95 |
| Total N (%) | 0.92 | 0.93 | 0.92 | 0.94 |
| P (%) | 0.94 | 0.89 | 0.92 | 0.87 |
| K (%) | 0.90 | 0.93 | 0.84 | 0.72 |
| Ca (%) | 0.98 | 0.95 | 0.83 | 0.81 |
| Mg (%) | 0.77 | 0.71 | 0.73 | 0.66 |
| Na (%) | 0.91 | 0.87 | 0.86 | 0.81 |
| S (%) | 0.98 | 0.91 | 0.93 | 0.83 |
| Mn (%) | 0.87 | 0.73 | 0.63 | NA |
| Fe (%) | 0.95 | 0.94 | 0.67 | 0.77 |
| Zn (ppm) | 0.95 | 0.94 | 0.00 | 0 |
| Cu (ppm) | 0.92 | 0.90 | 0.00 | 0 |
| Ni (ppm) | 0.92 | 0.97 | 0.00 | 0 |
| Cd (ppm) | 0.99 | 0.97 | 0.24 | 0 |
| Pb (ppm) | 0.95 | 0.98 | 0.00 | 0 |

For the calibrations, 67% of all organic amendment (OA) samples and Forest or XGBoost models were used.

rapidly quantify the concentrations of macro- and micronutrient elements present in the OA samples. Both MIR and pXRF had generally good agreement in calibrations for light elements (carbon, nitrogen) and holistic measures of OA quality (ash content), whereas pXRF tended to perform better for most heavier elements, though performance was nearly equal for P and S (Table 2). Forest regressions provided comparable results for MIR and XRF for Mg (Table 3), while Mn had weaker cross-validation results. The general pattern observed for MIR was that it generally performed well on elements that were not transition metals; with the exception being Fe, likely due to its higher abundance.

These results are novel for two reasons. First, XRF is unable to measure elements such as carbon or nitrogen directly and yet this study obtained excellent results ($R^2 > 0.86$) with XRF for both elements in both the cross-validation and hold-out sample sets (Tables 2 and 3). MIR is typically unable to measure elements with an atomic number $> 11$, such as P and K very well and yet we obtained reasonable results ($R^2 > 0.72$) for P in both the calibrations and validation sample sets and acceptable results ($R^2 > 0.30$) for K (Tables 2 and 3). We hypothesized that, because the matrix of the OA tested here is a mixture of organic components and silicates such as clays (S1 Fig in S1 File), there is a necessary inverse correlation between ash and carbon content. For pXRF analysis, there is also a change in density, as ash may have three times the density of organic material. Scattering of X-rays in this material will lead to different count rates reaching the detector from non-diagnostic portions of the unfiltered spectrum (7–10 keV) (Fig 5). This, coupled with the K-alpha line for silicon, allows for the strong predictive power for ash and N content because of density differences between them. As such, the success in identifying elements such as N (Figs 5 and 6) with portable XRF is understandable in this specific context, but likely not generalizable to plants and soils (the latter can also contain carbonates). However, the principle of narrowly defining a sample matrix to use machine learning techniques will likely produce future advances in the calibration of both x-ray and infrared data, as demonstrated here for OA.

Randomized cross validation trials (using 33% of all organic amendment samples and Forest or XGBoost models) confirmed the good predictive value of the XRF and MIR calibrations

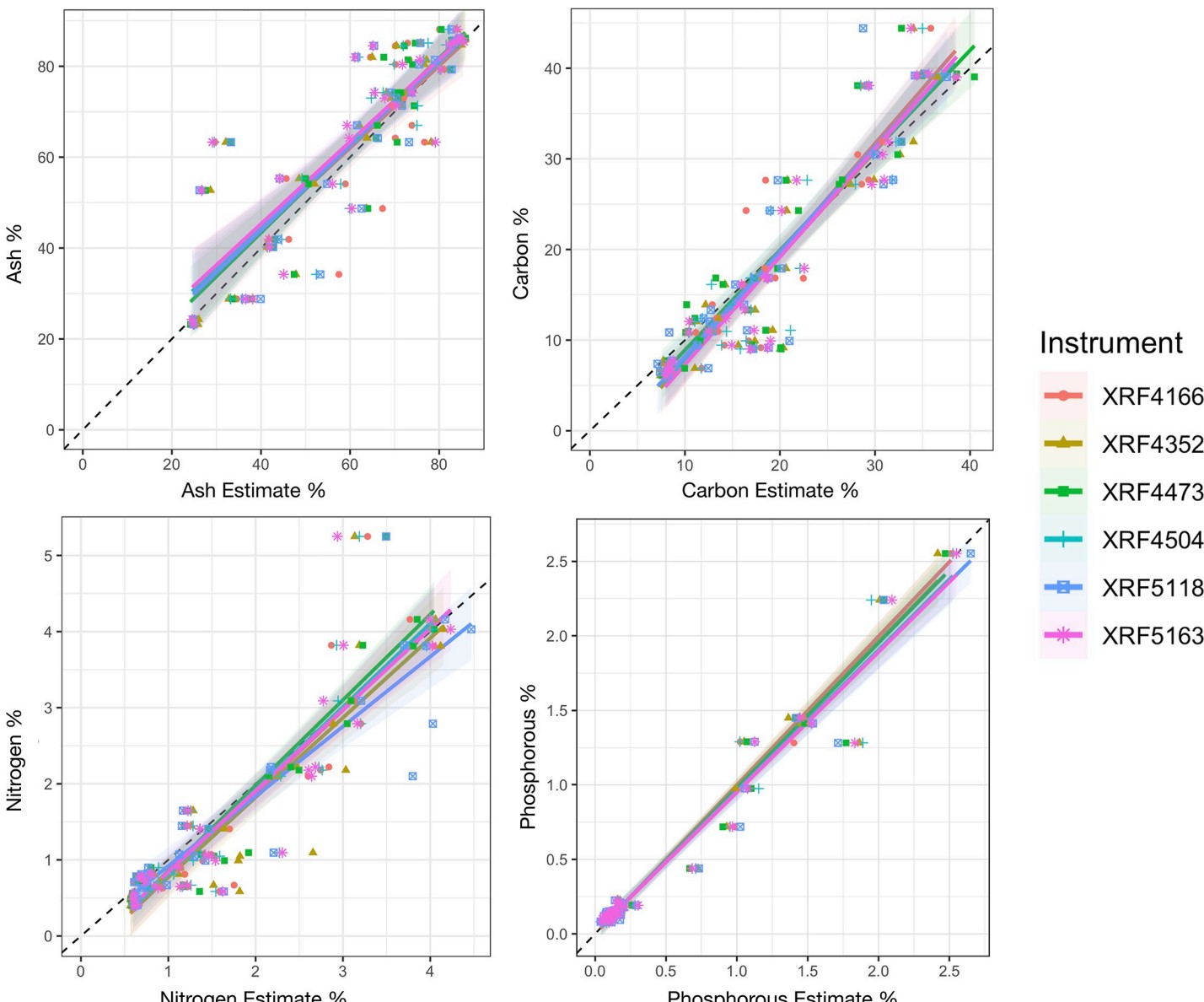

**Fig 6. Randomized cross-validation of 6 pXRF instruments for ash, total carbon, total nitrogen and total phosphorus.** For each characteristic, 33% of standards were withheld and treated as unknowns. The dotted line indicates the expected 1:1 ratio for pXRF estimates and known values and the shaded areas around all lines are 95% confidence predictions based on the calibration.

for most elements/characteristics in the hold-out validation set (Table 3; Figs 6 and 7). However, the cross validations also show that calibration model performance should be evaluated critically. For example, MIR calibration with XGBoost for Ca provides an $R^2$ of 0.81 (Table 2), while the average of 100 cross validation trials of the hold-out samples provides an $R^2$ of only 0.46 (Table 3). This suggests one of two possibilities: either XGBoosting is resulting in an over-fit of the data, or there is a critical threshold of the number of standards needed to provide an estimate of Ca that is met by the full dataset (n = 98) but not the partial (n = 65). Either way, the results show that randomized cross validation and hold-out validations are essential to interpret model accuracy and reliability. Our results indicate that MIR + machine learning is not yet a proven method to infer Mg and Mn concentrations in OA, contradicting the study of

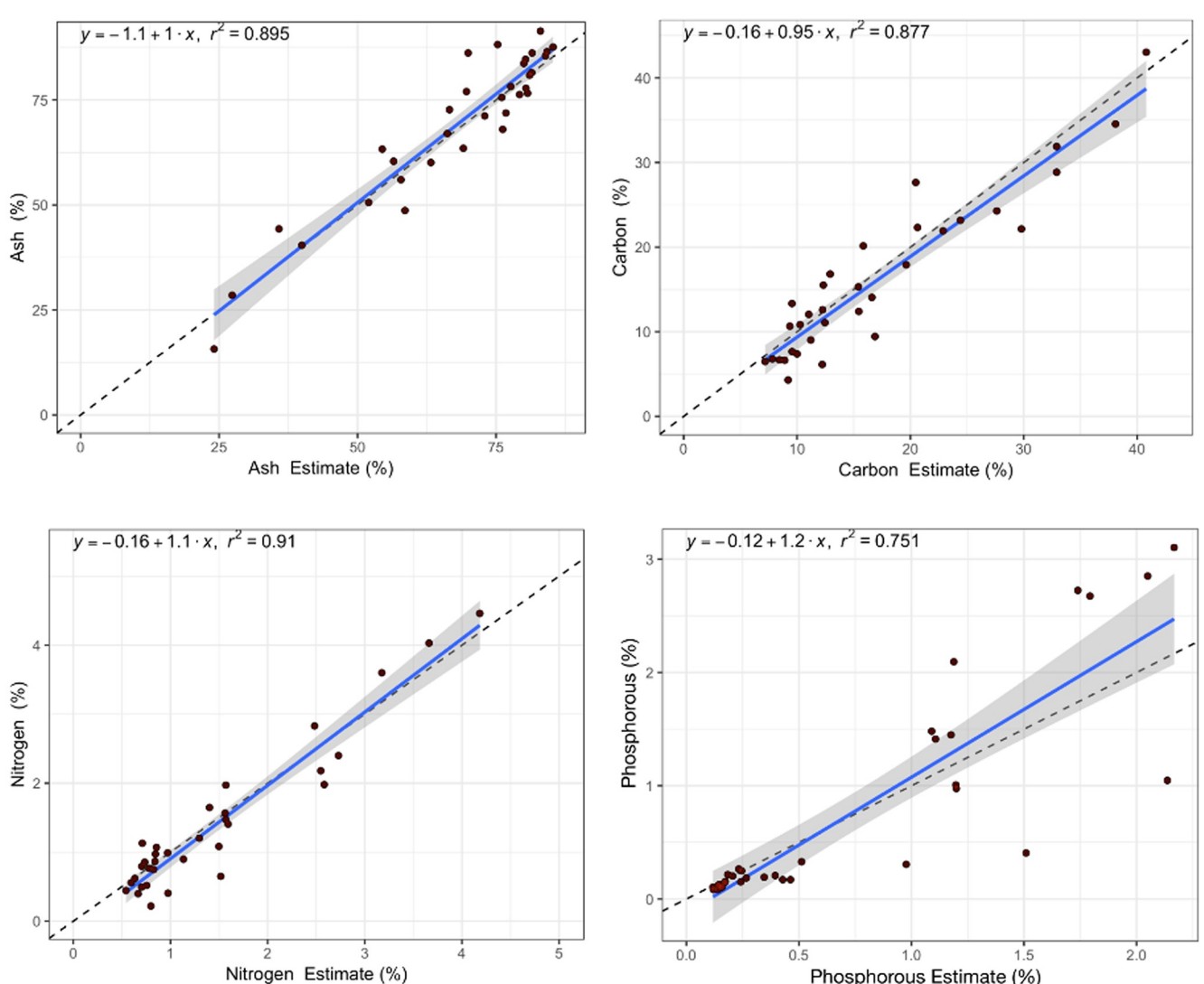

**Fig 7. Randomized cross-validation of DRIFT-MIR predictions for ash, total carbon, total nitrogen and total phosphorus.** For each element, 33% of standards were withheld and treated as unknowns. The dotted line indicates the expected 1:1 ratio for DRIFT-MIR estimates and known values. The shaded areas around all lines are 95% confidence predictions based on the calibration.

López-Núñez et al. [36] who claimed good calibrations for pXRF and a wide range of nutrients and contaminants in very similar OA (but the study did not include validations). MIR did not produce useable models for trace elements (Ni, Cu, Zn) or contaminants (Cd, Pb), while both machine learning model types produced useable models from XRF data; this is an unsurprising conclusion as these elements all have fluorescence peaks identifiable to low levels with XRF.

Based on theoretical considerations, MIR should be best for light elements like total C and total N while XRF should be best for evaluating elements like S and P. However, our results indicate that MIR came close to the performance of XRF on elements like P and S, while XRF came close to MIR in estimating total C and N (Table 3, Figs 6 and 7). Both devices had difficulty measuring Mg but surprisingly did better for P (Table 3). Elements which tend to correlate with clay content Na and K are elements which tend to correlate with the ash content, so the higher performance for these elements may be related to the success in identifying ash

**Table 3. Correlation coefficients ($R^2$) and the slope of the regression line between observed and predicted values for all cross-validation trials with pXRF and MIR and all sample characteristics measured in the hold-out validation.**

| | pXRF Forest $R^2$ | pXRF Forest Slope | pXRF XGBR$^2$ | pXRF XGB Slope | MIR Forest $R^2$ | MIR Forest Slope | MIR XGB $R^2$ | MIR XGB Slope | Best Method | Best Model |
|---|---|---|---|---|---|---|---|---|---|---|
| Ash (%) | 0.83 | 1.01 | 0.83 | 1.01 | 0.89 | 1.10 | 0.86 | 1.03 | MIR | XGBoost |
| Total C (%) | 0.88 | 1.03 | 0.87 | 1.01 | 0.90 | 1.11 | 0.89 | 1.01 | MIR | XGBoost |
| Total N (%) | 0.86 | 1.00 | 0.86 | 1.01 | 0.87 | 1.15 | 0.83 | 1.02 | pXRF | Forest |
| P (%) | 0.66 | 0.96 | 0.64 | 1.06 | 0.77 | 1.18 | 0.60 | 0.89 | pXRF | Forest |
| K (%) | 0.78 | 0.83 | 0.79 | 0.79 | 0.47 | 1.06 | 0.30 | 0.66 | pXRF | Forest |
| Ca (%) | 0.66 | 0.98 | 0.63 | 0.81 | 0.69 | 1.52 | 0.46 | 0.68 | pXRF | Forest |
| Mg (%) | 0.42 | 0.78 | 0.43 | 0.76 | 0.27 | 1.24 | 0.38 | 0.52 | pXRF | XGBoost |
| Na (%) | 0.65 | 0.97 | 0.63 | 0.89 | 0.65 | 1.18 | 0.43 | 0.62 | pXRF | Forest |
| S (%) | 0.70 | 1.16 | 0.68 | 0.99 | 0.88 | 1.11 | 0.73 | 0.90 | pXRF | XGBoost |
| Mn (%) | 0.25 | 0.77 | 0.25 | 0.71 | 0.08 | 0.77 | NA | NA | pXRF | Forest |
| Fe (%) | 0.89 | 1.05 | 0.88 | 1.05 | 0.28 | 1.07 | 0.19 | 0.68 | pXRF | Forest |
| Zn (ppm) | 0.45 | 0.83 | 0.43 | 0.76 | NA | NA | NA | NA | pXRF | Forest |
| Cu (ppm) | 0.83 | 1.12 | 0.62 | 0.81 | NA | NA | NA | NA | pXRF | Forest |
| Ni (ppm) | 0.61 | 1.02 | 0.59 | 0.90 | NA | NA | NA | NA | pXRF | Forest |
| Cd (ppm) | NA | NA | NA | NA | NA | NA | NA | NA | NA | NA |
| Pb (ppm) | 0.84 | 1.01 | 0.81 | 0.99 | NA | NA | NA | NA | pXRF | Forest |

The cross-validation trials used 33% of all organic amendment (OA) samples and Forest or XGBoost models.

content. With few exceptions, MIR and XRF were relatively interchangeable techniques for estimating all properties analyzed in this wide range of OA.

## Conclusions

We conclude that combining MIR and XRF spectral methods with machine learning techniques enables rapid, portable, and nondestructive measurement of a full suite of nutrients in OA on both devices independently. The approach is also scalable, as the calibration process for XRF can be at least partially automated provided each new instrument is calibrated against common standards. These results are significant in that:

1. XRF is capable of estimating properties like carbon and nitrogen content of OA;

2. MIR is capable of estimating P and S in OA as well;

3. Portable non-destructive spectrometry paired with machine learning can provide a comprehensive nutrient profile with minimal sample preparation outside a traditional laboratory environment;

4. XRF allows contaminants to be detected, e.g. the presence of trace amounts of potentially toxic metals like e.g. Zn, Cu and Ni–there were good calibration/validations in our study for these elements.

If there is one key scope of this work, it is that smallholders need good returns on their investments, especially OA, mostly in terms of increased yields to achieve greater profit and/or food security. Therefore, portable MIR and XRF spectrometers in conjunction with machine learning are adequate solutions to support nutrient management with minimal cost for analysis per sample. For spectrometry at large, machine learning techniques can extract more

actionable information than has been previously recognized. To our knowledge, this is the first study to compare the performance of pXRF and FTIR as rapid analytical methods for the determination and monitoring of major and trace nutrient elements in OA. Both methods performed well for most parameters analyzed but pXRF did slightly better for heavier elements whereas FTIR was superior for light elements and the ash content. The combination with machine learning helps to reduce uncertainties in assessing OA quality and, hence, enables better decision-making especially for comprehensive nutrient management for all types of farms. It also allows to identify and avoid contaminated OA fertilizers, thereby protecting soils and the environment from pollution. Future work will now evaluate the use of XRF for testing of conventional mineral fertilizers.

## Supporting information

**S1 File.**
(DOCX)

## Acknowledgments

This work received support from the World Agroforestry (ICRAF) Soil-Plant Spectral Diagnostics Laboratory and the Rothamsted Research, Department of Sustainable Agriculture Sciences staff who were involved in the collection, management and analysis of the organic amendment samples.

## Author Contributions

**Conceptualization:** Erick K. Towett, Lee B. Drake, Stephan M. Haefele, Steve P. McGrath, Keith D. Shepherd.

**Data curation:** Erick K. Towett, Lee B. Drake, Gifty E. Acquah.

**Formal analysis:** Erick K. Towett, Lee B. Drake, Gifty E. Acquah.

**Funding acquisition:** Steve P. McGrath, Keith D. Shepherd.

**Investigation:** Erick K. Towett, Lee B. Drake, Gifty E. Acquah.

**Methodology:** Erick K. Towett, Lee B. Drake, Gifty E. Acquah, Stephan M. Haefele.

**Project administration:** Steve P. McGrath, Keith D. Shepherd.

**Resources:** Steve P. McGrath, Keith D. Shepherd.

**Software:** Lee B. Drake.

**Supervision:** Steve P. McGrath, Keith D. Shepherd.

**Validation:** Erick K. Towett, Lee B. Drake, Stephan M. Haefele.

**Visualization:** Erick K. Towett, Lee B. Drake.

**Writing – original draft:** Erick K. Towett, Lee B. Drake, Stephan M. Haefele.

**Writing – review & editing:** Gifty E. Acquah, Stephan M. Haefele, Steve P. McGrath, Keith D. Shepherd.

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
