## [Decision Letter · Decision Letter 0]

8 Sep 2020

PONE-D-20-15361

Comprehensive Nutrient Analysis in Agricultural Organic Amendments Through Non-Destructive Assays Using Machine Learning

PLOS ONE

Dear Dr. Erick K Towett,

Thank you for submitting your manuscript to PLOS ONE. After careful consideration, we feel that it has merit but does not fully meet PLOS ONE’s publication criteria as it currently stands. Therefore, we invite you to submit a revised version of the manuscript that addresses the points raised during the review process.

We look forward to receiving your revised manuscript.

Kind regards,

Balasubramani Ravindran, Ph.D

Academic Editor

PLOS ONE

Journal Requirements:

Reviewers' comments:

Reviewer's Responses to Questions

**Comments to the Author**

1. Is the manuscript technically sound, and do the data support the conclusions?

Reviewer #1: Yes

Reviewer #2: Yes

Reviewer #3: Yes

2. Has the statistical analysis been performed appropriately and rigorously? 

Reviewer #1: No

Reviewer #2: Yes

Reviewer #3: Yes

3. Have the authors made all data underlying the findings in their manuscript fully available?

Reviewer #1: Yes

Reviewer #2: Yes

Reviewer #3: Yes

4. Is the manuscript presented in an intelligible fashion and written in standard English?

Reviewer #1: Yes

Reviewer #2: Yes

Reviewer #3: Yes

5. Review Comments to the Author

Reviewer #1: THE STUDY VALIDATE THE APPLICATION OF XPS FOR THE REAL-TIME SOIL NUTRIENT ANALYSIS.

IT IS WELL ACCOMPLISHED WITH NECESSARY METHODS.

the conclusion is supported with the necessary results.

authors approach to use mechine learning for the data analysis is to be appreciated.

Reviewer #2: General commends

The author did good piece of work on machine learning helps to find percentage of carbon and nitrogen presence in the soil. XRF is capable of estimating properties like carbon and nitrogen. the work on artificial neural networks was well explained.

The author addresses following question

1. In conclusion part authors discussing reason it may be avoid and rewrite.

2. From your study, How MIR + machine learning useful to derive finding of Mg and Mn concentrations?

3. Mentation accuracy difference (Percentage) Tracer 5i pXRF 900F4166 and 900F4473 instrument.

4. Line number 211, calibrations, 67% was chosen randomly what could be the reason for choosing 67%. If more 80 % and above does the values change?

Reviewer #3: Comments to the Authors

The authors have tried to find alternative sources for Comprehensive Nutrient Analysis in Agricultural Organic Amendments through Non-Destructive Assays Using Machine Learning. The work done by the authors is commendable and applaudable. It is considered commendable since this study provides an alternative solution for portable spectrometry in combination with machine learning a scalable solution to provide comprehensive nutrient analysis for organic amendments.

Abstract

Avoid abbreviations in the abstract

Introduction

The authors have arranged the literatures and context relating the significance of choosing the objective and scope of the study. The following comments in this section are

Line 26 - Small-scale and family farmers – Define.

Line 27 – Authors are to include some literature related to food supply in developing countries. Readers will be interested to understand about what is food supply in developing countries through some literatures in introduction section.

Line 109 – The authors are recommended to add certain background studies related to macro and micronutrients. Further, additional details for C and N, as well as ash in this study are also encouraged. This information will act as the state of the art for the readers and also compare with the current research findings of the authors.

Materials and Methods

The authors have presented the methodology in a standard and technical aspects. However there are certain facts that are to be improved and included to increase the reader’s interest.

Line 176 – Though a detailed explanation on experimental model has been provided. Authors are encouraged to provide real time pictures of the experimental study carried out to increase the reader’s interest (If available). In addition, an image of the field used in this study can be included in the manuscript.

Results

This section has been explained in appropriate manner.

Discussion

This section has been discussed in detail by the authors and the literatures stated in this section have been covered well related to the objective and scope of this study.

Conclusion

Include solid findings with quantifiable results. Add the scope and future directions in brief.

General comments to authors

I encourage and recommend the authors to also provide site images, to increase the curiosity in readers. I would like to recommend minor revision of this study and accept this manuscript in its present form.

6. PLOS authors have the option to publish the peer review history of their article (what does this mean?). If published, this will include your full peer review and any attached files.

Reviewer #1: No

Reviewer #2: No

Reviewer #3: No

---

## [Author Response · Author response to Decision Letter 0]

21 Sep 2020

Dear Balasubramani Ravindran, 

Academic Editor

PLOS ONE

Subject: Manuscript Revision Submission

Ms. Ref. No.: PONE-D-20-15361

Title: Comprehensive Nutrient Analysis in Agricultural Organic Amendments Through Non-Destructive Assays Using Machine Learning

Journal: PLOS ONE

We hereby submit a revised version of our manuscript titled: " Comprehensive nutrient analysis in agricultural organic amendments through non-destructive assays using machine learning" by Towett K. Erick, Drake B. Lee, Acquah E. Gifty, Haefele M. Stephan McGrath P. Steve, and Shepherd D. Keith, to be considered for publication in PLOS ONE.

Thank you very much for the opportunity to revise our manuscript to address the reviewers’ comments. We have carefully undertaken the revision of the manuscript in light of the peer reviewers 2 and 3 comments geared towards making the MS more informative. WE have also addressed the editors comments. Copies of the manuscript, Tables and Figures with tracks the changes are submitted separately as requested. The responses to the specific comments by the reviewers are given in a separate submitted file named “response to reviewers”.

The responses to the specific comments by the reviewers are given as follows.

Response to Reviewers' Comments

Reviewer's Responses to Questions

Comments to the Author

1. Is the manuscript technically sound, and do the data support the conclusions?

Reviewer #1: Yes

Reviewer #2: Yes

Reviewer #3: Yes

2. Has the statistical analysis been performed appropriately and rigorously? 

Reviewer #1: No

Reviewer #2: Yes

Reviewer #3: Yes

3. Have the authors made all data underlying the findings in their manuscript fully available?

Reviewer #1: Yes

Reviewer #2: Yes

Reviewer #3: Yes

4. Is the manuscript presented in an intelligible fashion and written in standard English?

Reviewer #1: Yes

Reviewer #2: Yes

Reviewer #3: Yes

5. Review Comments to the Author

Reviewer #1: 

THE STUDY VALIDATE THE APPLICATION OF XPS FOR THE REAL-TIME SOIL NUTRIENT ANALYSIS.

IT IS WELL ACCOMPLISHED WITH NECESSARY METHODS.

the conclusion is supported with the necessary results.

authors approach to use mechine learning for the data analysis is to be appreciated.

Response to Comment: We very much appreciate the reviewer’s recommendation and constructive suggestions on how this MS may be revised to be more informative.

Reviewer #2: General commends

The author did good piece of work on machine learning helps to find percentage of carbon and nitrogen presence in the soil. XRF is capable of estimating properties like carbon and nitrogen. the work on artificial neural networks was well explained.

The author addresses following question

1. In conclusion part authors discussing reason it may be avoid and rewrite.

Response to comment no. 1: We agree that the conclusions of the manuscript need not to have reasons and we have reworded it from the initial wording “These results are significant for the following reasons:” and we have now rewritten it to read “ These results are significant in that:…..”.

2. From your study, How MIR + machine learning useful to derive finding of Mg and Mn concentrations?

Response to comment no. 2: When we revised the manuscript, we considered the concern raised and have edited the discussion section to include a statement that “Forest regressions provided comparable results for MIR and XRF for Mg (Table 3), while Mn had weaker cross-validation results. That said, the general pattern observed for MIR was that it generally performed well on elements that were not transition metals; with the exception being Fe, likely due to its higher abundance” in lines 342-345. 

3. Mentation accuracy difference (Percentage) Tracer 5i pXRF 900F4166 and 900F4473 instrument.

Response to comment no. 3. The difference between 900F4166 and 900F4473 isn’t accuracy, but rather photon flux - the XRF tube in 900F4473 generated more x-rays per unit time for the energy range of light elements (1 - 4 keV); this could be due to any number of manufacturing differences, such as the thickness of the Rh anode in the tube, the thickness of the Be window used on either the tube or the detector, or small differences in geometry that improve fluorescence in a given energy range. It is precisely these small manufacturing differences that produce the need for individual instrument calibrations. This is described at the end of the second paragraph on page 9, lines 327 to 333. 

4. Line number 211, calibrations, 67% was chosen randomly what could be the reason for choosing 67%. If more 80 % and above does the values change?

Response to comment no. 4: Random sampling is the simplest way of selecting samples as it creates a subset that follows the statistical distribution of the original dataset. The actual value used for cross-validation is ultimately about tradeoffs - what is the minimal number of samples needed to train a model to generalize to a larger data set? A 20% validation split (e.g. 80% of samples used for training) would produce a higher R2 and slope closer to 1 for both the training and test data sets. The more restrictive 33% cross-validation sample (e.g. 67% used for training data) was used given the significance of the claims - accurately quantifying N or C using XRF (and P or S using MIR) is a notable expansion of each instrument’s capability. The cross-validation split was used to stress-test these claims to demonstrate the generalizability of the models, rather than get the highest possible R2 or slope closest to 1. It would be interesting to rerun these data at a 20% split - but it would take at least two weeks and thus extended beyond the review time (as well as be a costly allocation of computer resources). We have added text to clarify the rational behind this particular cross-validation split choice in lines 231-234. 

Randomized cross-validation in general is an unbiased method, it is also efficient as more samples are required to achieve the representativeness of the data. This method is commonly used as it is easy to carry out, and unbiased. Shepherd & Walsh (2002) and McCarty et al. (2002) are some exemplar studies where this sampling approach has been utilized in spectral modelling studies where each dataset is first randomly split into calibration and validation set (75% and 25% respectively). In general, although an increase in calibration set size could increase the performance of the model (at thus increasing the calibration set to 80% would increase the accuracy of the validation data), it has been found that in many datasets, calibration sample size >75% does not provide much improvement to model prediction and also means that only 25% of the samples need to be fully analysed to provide a good calibration set. Following this approach, for our study we chose randomly 67% of the samples for validation and 33% for calibration as the dataset was split based on the unique reference properties and based on sample types (Table 1 & Figure 2).

Shepherd KD, Walsh MG. 2002. Development of reflectance spectral libraries for characterization of soil properties. Soil Science Society of America Journal 66:988_998. DOI 10.2136/sssaj2002.9880.

McCarty GW, Reeves JB, Reeves VB, Follett RF, Kimble JM. 2002. Mid-infrared and near-infrared diffuse reflectance spectroscopy for soil carbon measurement. Soil Science Society of America Journal 66:640_646 DOI 10.2136/sssaj2002.6400.

Reviewer #3: Comments to the Authors

1). The authors have tried to find alternative sources for Comprehensive Nutrient Analysis in Agricultural Organic Amendments through Non-Destructive Assays Using Machine Learning. The work done by the authors is commendable and applaudable. It is considered commendable since this study provides an alternative solution for portable spectrometry in combination with machine learning a scalable solution to provide comprehensive nutrient analysis for organic amendments.

1). Abstract

Avoid abbreviations in the abstract

Response to comment no. 1: Many thanks for this comment. The abbreviations have been corrected for the elements Carbon and Nitrogen.

2). Introduction

The authors have arranged the literatures and context relating the significance of choosing the objective and scope of the study. The following comments in this section are

Line 26 - Small-scale and family farmers – Define.

Response to comment no. 2: Thanks for the comments. This reference to the family farms has been deleted. Although it can be argued that limited areas under smallholder farming systems are closer to the homestead, family farms are those small plots nearing the farmhouse where most recycling of organic waste occurs and a diverse range of crops is produced.

3). Line 27 – Authors are to include some literature related to food supply in developing countries. Readers will be interested to understand about what is food supply in developing countries through some literatures in introduction section.

Response to comment no. 3: Thank you for this comment. This would be too broad for the present scope of the manuscript and out of context with the study. However, a literature on the above mentioned food supply dynamics in Sub-Saharan Africa countries has been included in lines 32-43 with particular emphasis on the need for establishing effective quality assurance mechanisms.

4). Line 109 – The authors are recommended to add certain background studies related to macro and micronutrients. Further, additional details for C and N, as well as ash in this study are also encouraged. This information will act as the state of the art for the readers and also compare with the current research findings of the authors.

Response to comment no. 4: The reviewer refers here to the last line of objectives, and it would be rather unconventional to add references here. However, the requested descriptions and references are available in the text line 99 to 104 and 108 to 130. 

5). Materials and Methods

The authors have presented the methodology in a standard and technical aspects. However there are certain facts that are to be improved and included to increase the reader’s interest.

Response to comment no. 5: This is a very general comment which is difficult to address. We do believe that we introduced the method from a technical point of view but also in relation to its possible uses, so we thought this should create interest for the paper. 

6). Line 176 – Though a detailed explanation on experimental model has been provided. Authors are encouraged to provide real time pictures of the experimental study carried out to increase the reader’s interest (If available). In addition, an image of the field used in this study can be included in the manuscript.

Response to comment no. 6: We could include images of the equipment in action, taken in the laboratory or to simulate field as we did not take measurements in the field for this study. We would ask the editor to decide if such images are something the journal would like to include - we have included S1 – S3 Photos after the S1 and S2 Figures in the supporting information for consideration. 

7). Results

This section has been explained in appropriate manner.

Response to comment no. 7: We very much appreciate the reviewer comment.

8). Discussion

This section has been discussed in detail by the authors and the literatures stated in this section have been covered well related to the objective and scope of this study.

Response to comment no. 8: Thanks for this comment.

9). Conclusion

Include solid findings with quantifiable results. Add the scope and future directions in brief.

Response to comment no. 9: We also very much appreciate the reviewer’s constructive suggestions and recommendations to add the scope and future directions to the conclusion. This has been done at the end of the conclusion, lines 412-414 and 429-431. 

10). General comments to authors

I encourage and recommend the authors to also provide site images, to increase the curiosity in readers. I would like to recommend minor revision of this study and accept this manuscript in its present form.

Response to comment no. 10: We also very much appreciate the reviewer’s constructive suggestions and recommendations on including images (S1 – S3 Photos) in the supporting information to manuscript to be more informative and we have undertaken to include the same as supplementary to the MS and let the editor decide on whether these can be included, as outlined in the response to reviewer question 6 above.

---

## [Decision Letter · Decision Letter 1]

10 Nov 2020

Comprehensive nutrient analysis in agricultural organic amendments through non-destructive assays using machine learning

PONE-D-20-15361R1

Dear Dr. Erick K Towett ,

We’re pleased to inform you that your manuscript has been judged scientifically suitable for publication and will be formally accepted for publication once it meets all outstanding technical requirements.

Kind regards,

Balasubramani Ravindran, Ph.D

Academic Editor

PLOS ONE

Reviewers' comments:

Reviewer's Responses to Questions

**Comments to the Author**

1. If the authors have adequately addressed your comments raised in a previous round of review and you feel that this manuscript is now acceptable for publication, you may indicate that here to bypass the “Comments to the Author” section, enter your conflict of interest statement in the “Confidential to Editor” section, and submit your "Accept" recommendation.

Reviewer #1: All comments have been addressed

Reviewer #2: All comments have been addressed

Reviewer #3: All comments have been addressed

2. Is the manuscript technically sound, and do the data support the conclusions?

Reviewer #1: Yes

Reviewer #2: Partly

Reviewer #3: Yes

3. Has the statistical analysis been performed appropriately and rigorously? 

Reviewer #1: (No Response)

Reviewer #2: Yes

Reviewer #3: Yes

4. Have the authors made all data underlying the findings in their manuscript fully available?

Reviewer #1: Yes

Reviewer #2: Yes

Reviewer #3: Yes

5. Is the manuscript presented in an intelligible fashion and written in standard English?

Reviewer #1: Yes

Reviewer #2: Yes

Reviewer #3: Yes

6. Review Comments to the Author

Reviewer #1: (No Response)

Reviewer #2: The Authors answers all the question hence we can proceed for publication.

Reviewer #3: (No Response)

7. PLOS authors have the option to publish the peer review history of their article (what does this mean?). If published, this will include your full peer review and any attached files.

Reviewer #1: No

Reviewer #2: **Yes: **Dr G. RAMALINGAM

Reviewer #3: No

---

## [Editor Report · Acceptance letter]

1 Dec 2020

PONE-D-20-15361R1 

Comprehensive nutrient analysis in agricultural organic amendments through non-destructive assays using machine learning 

Dear Dr. Towett:

I'm pleased to inform you that your manuscript has been deemed suitable for publication in PLOS ONE. Congratulations! Your manuscript is now with our production department. 

Kind regards, 

on behalf of

Dr. Balasubramani Ravindran 

Academic Editor

PLOS ONE